# Synthesis and Characterization of High-Performing Sulfur-Free Tannin Foams

**DOI:** 10.3390/polym12030564

**Published:** 2020-03-04

**Authors:** Jonas Eckardt, Jonas Neubauer, Thomas Sepperer, Sandro Donato, Michela Zanetti, Nicola Cefarin, Lisa Vaccari, Marcel Lippert, Matthias Wind, Thomas Schnabel, Alexander Petutschnigg, Gianluca Tondi

**Affiliations:** 1Forest Products Technology & Timber Constructions Department, Salzburg University of Applied Sciences, Marktstrasse 136a, 5431 Kuchl, Austria; jonas.eckardt@fh-salzburg.ac.at (J.E.); jonas.neubauer@fh-salzburg.ac.at (J.N.); thomas.sepperer@fh-salzburg.ac.at (T.S.); mlippert.htw-m2019@fh-salzburg.ac.at (M.L.); mwind.htw-m2019@fh-salzburg.ac.at (M.W.); thomas.schnabel@fh-salzburg.ac.at (T.S.); alexander.petutschnigg@fh-salzburg.ac.at (A.P.); 2Salzburg Center for Smart Materials, Jakob-Haringer-Strasse 2a, 5020 Salzburg, Austria; 3Physics Department, University of Calabria, Via Pietro Bucci, cubo 31C, 87036 Rende (CS), Italy; sandro.donato@fis.unical.it; 4Elettra-Sincrotrone Trieste S.C.p.A., Strada Statale 14-km 163,5 in AREA Science Park, 34149 Trieste, Basovizza, Italy; nicola.cefarin@elettra.eu (N.C.); lisa.vaccari@elettra.eu (L.V.); 5Land, Environment, Agriculture and Forestry Department, University of Padua, Viale dell’Università 16, 35020 Legnaro (PD), Italy; michela.zanetti@unipd.it

**Keywords:** polyphenols, flavonoids, furanics, sulphur, bio-polymers, pentane, heating value

## Abstract

Tannin foams are green lightweight materials that have attracted industrial interest for the manufacturing of sandwich panels for insulation purposes. However, the dimensions of the cells and the presence of sulfur in the formulation developed until now have discouraged their upscaling. In this work, we present the synthesis and the characterization of the more promising small cell and sulfur-free materials. It was observed that, with respect to standard ones, foams catalyzed with nitric acid present similar physical properties and more phenolic character, which favors the absorption of ionic pollutants. Conversely, the foams blown with aliphatic solvents and surfactants present smaller pores, and higher mechanical and insulating properties, without affecting the chemical properties or the heating value. The combined foam produced with nitric acid as a catalyst and petroleum ether as a blowing agent result in sulfur-free and small cell material with overall improved features. These foams have been produced at 30 × 30 × 3 cm^3^, with high homogeneity and, to date, they represent the most suitable formulation for industrial upscaling.

## 1. Introduction

The most commonly applied foams for the insulation of buildings in the European market are made of polystyrene or polyurethane [1,2,3]. Even if, nowadays, the interest in sustainable solutions is growing, the alternative foams under investigation are aerogels or bio-based polyurethanes [4,5]. Nevertheless, the tannin foams have been constantly monitored for a decade now, for their structure and morphology [6,7,8], and their chemical and physical properties [9,10,11]. Tannin foams have shown remarkable properties: they are based on natural resources [12,13]; they are very light, and can be produced with different heating systems [14,15]; they are fire and water resistant [16]; they can be modified and functionalized [17,18,19]; they can be easily combined with different top layers [20]; and they can even be reused at the end of their working life [21].

Despite the upscaling projects performed [22], there are still some technological drawbacks to be considered: (i) the consistent presence of sulfur in the formulation due to a sulfur-based catalyst (traces of sulfites due to the extraction process could still be present), and (ii) the morphological inhomogeneity occurring during the upscaling process, which affects the properties of the foams. For filling the remaining gap between the material properties and the industrial requirements, in the present work we aim the overcome these drawbacks by using alternative formulations.

## 2. Materials and Methods

### 2.1. Materials

The extract Weibull AQ, provided by the company Tanac (Montenegro, RS, Brazil), and furfuryl alcohol from Transfurans Chemicals (Geel, Belgium), were the basis for every formulation. The blowing agents used were diethylether (DEE), pentane (Pent), and petroleum ether (PE), while the catalysts were sulfuric acid (SA) and nitric acid (NA) from Roth (Karlsruhe, Germany). The surfactant Tween80 and the water pollutants methylene blue and sodium dodecyl sulfate (SDS) were purchased from VWR (Darmstadt, Germany).

### 2.2. Synthesis of the Tannin Foams

Tannin powder, furfuryl alcohol, water, surfactant, and blowing agent were mixed until a highly homogeneous emulsion was obtained with the proportions reported in Table 1. Then, the catalyst was added and vigorously stirred for 30 s. The formulation was poured into a preheated Medium Density Fiberboard (MDF) mold, and this was laid into a Höfer (Taiskirchen, Austria) press at 90 °C. The foams were blown and hardened in 5 min, and then they were removed from the mold and conditioned at 20 °C and 65% relative humidity for one week before testing.

### 2.3. Intrinsic Physical Characterization

#### 2.3.1. Bulk Density, Skeletal Density, and Porosity

The bulk density (d) was calculated by measuring the length (l), width (w), height (h), and weight (m) for each cube, according to Equation (1).
d = m/(l × w × h)(1)

The skeletal density of the different formulations was measured by He-pycnometry, using the gas pycnometer Quantachrome Instrument (Boynton Beach, FL, USA) Ultrapyc 1200e, and the porosity (P) of the material was calculated with Equation (2), where d is the bulk density and d_s_ is the skeletal density.
P = 1 − (d/d_s_)(2)

#### 2.3.2. Cell Dimensions and Orthotropicity

The cell size dimensions were measured with a reflected light microscope (Nikon SMZ 1500, Tokio, Japan). For each formulation, at least 50 measurements in length and 100 in width were carried out in the direction parallel to the growth on three samples. The average foam cell diameter (Av.D) has been calculated with Equation (3), where D is the average of 150 measurements.
Av.D = (π/4) × D(3)

Orthotropicity (o) was calculated by using Equation (4), where Av.l is the average length and Av.w is the average width.
o = Av.l/Av.w(4)

#### 2.3.3. X-ray Tomography Acquisitions

From the center of each foam block, a cube of 3 × 3 × 3 cm^3^ was extracted. Those samples were imaged at the Tomolab laboratory, at the synchrotron facility of Elettra Sincrotrone Trieste.

The X-ray source was a sealed microfocus Hamamatsu X-ray tube (model L12161-07) with a Tungsten target, which can operate at voltages between 40 and 150 KV and at a maximum power of 75 W. The tube voltage was set to 40 keV and the current to 100 μA. The focal spot size was set to 5 μm. The detecting system consisted of a Hamamatsu CMOS camera (C12849-102, Hamamatsu, Japan) with a pixelsize of 6.5 μm and an active area of 20.48^2^ pixels^2^. The sample-to-detector distance was 120 mm, while the source-to-detector distance was 370 mm. Considering the magnification factor ~3, the equivalent pixelsize of the acquired projection was equal to 2.1 μm, and the corresponding imaged volume was approximatively equal to 80 mm^3^. For each sample, 1800 equi-angular projections over 360°, with an exposure time of 2 s per projection, were acquired. Axial slices were reconstructed with an isotropic voxel size of 2.1 μm^3^, using the standard FDK algorithm [23] for cone-beam geometry. Prior to volume rendering, digital slices were processed with a 3D median filter algorithm for noise reduction. For each foam, a volume of interest of nearly 5 mm^3^ was selected for the 3D rendering done with Avizo® 9.3.

#### 2.3.4. Heating Value

The low heating value on a wet basis (PCI_w_), and high heating value on a dry basis (PCS_0_) of ground, tannin-based foam samples were calculated following the methodology described in the ISO standard 18,125 [24]. The high heating value of the samples as received has been measured using an IKA (Staufen, Germany) C-200 calorimeter. As indicated in the ISO standard, the average of two replications was calculated, and the repeatability limit did not differ by more than 140 J/g.

### 2.4. Intrinsic Chemical Characterization

#### Attenuated Total Reflectance Fourier-Transform Infrared (ATR FT-IR) Spectroscopy

The five foams were crushed into powder, dried, and finally scanned using a Perkin Elmer (Perkin-Elmer, Waltham, MA, USA) Frontier FT-IR spectrometer, equipped with an ATR Miracle accessory, with the same procedure applied in the previous work of the group [25]. Sixteen scans were performed for each sample, at a resolution of 4 cm^−1^, in the range of 4000 to 600 cm^−1^, and this was repeated in triplicate. Bio Rad KnowItAll (BioRad, California, CA, USA) software was used for normalizing and averaging the spectra.

### 2.5. Extrinsic Physical Characterization

#### 2.5.1. Compression Strength

The compression strength was measured with the universal testing machine Zwick Roell Z250 (Zwick-Roell, Ulm, Germany) on samples 3 × 3 × 3 cm^3^, according to the standard DIN 52,185 [26]. Nine specimens were tested parallel to the growth direction. The rate was set to 2 mm/min. If a clear break of the material appeared within 10% of specimen deformation, that value was taken as the result. If no clear break occurred within this range, the value at 10% of the deformation of the tested sample was taken as the maximum compression strength.

#### 2.5.2. Thermal Conductivity

The thermal conductivity was measured with a λ-Meter Lambda-Messtechnik EP500e (Dresden, Germany) device on samples with dimension 25 × 25 × 3 cm^3^. The measurements took place at three different temperatures (10, 25, and 40 °C) by applying 10K of temperature difference to the upper heating plate for each temperature.

#### 2.5.3. Fire Resistance

Conditioned cubes of 2 cm per side of foam were held on a lab-made device laying on an analytical balance. The samples were exposed to 3 min of direct conical blue-flame at a distance of 3 cm from the Bunsen burner, and the weight was registered every 30 s during the burning phase (3 min), and during the glowing phase (2 min), before self-extinguishing. Five samples for each formulation were tested, and the percentage of mass loss was presented as average.

### 2.6. Extrinsic Chemical Characterization

#### 2.6.1. Acid Recovery

The produced foams were pulverized and dried at 103 °C. Afterwards, 5 g of dry tannin-foam powder was added to 250 mL water and magnetically stirred for 3 min. The solution was then filtrated through a paper filter with a pore size of 12–25 µm and washed with a total amount of 750 mL of water. The pH-value of the solution was measured with a WTW inoLab pH 720 m (Xylem, New York, NY, USA). The process was done in triplicate.

#### 2.6.2. Pollutant Absorption

The pollutant adsorption capacity test was set up by exposing 4 mg of the leached tannin foam powders to 20 ppm solutions of: (i) methylene blue, and (ii) sodium dodecyl sulfate for 48 h under stirring conditions, and 24 h under stationary conditions, at room temperature and in a dark environment, following the procedure reported by other sources [21,27,28].

These solutions were filtered with a vacuum glass filter of 16–40 µm (Por.3), and then the intensity of the signal of the solutions was measured with a Shimadzu UVmini 1240 UV-VIS spectrophotometer (Shimadzu, Kyoto, Japan) at 656 nm.

## 3. Results

The five foams, subject of the present study, have been characterized for their intrinsic and extrinsic properties from a physical and chemical point of view. In Figure 1, the different appearance of the standard (SA-DE) and the petroleum ether (NA-PE) foams are reported.

### 3.1. Intrinsic Physical Properties

The intrinsic physical properties that have been considered for the foams are summarized in Table 2.

The measurements presented in this section are in line with the findings of previous studies of the group. Free expansion tannin foams reached density values down to 23 kg/m^3^ [29]; porosities of 92.5% to 96.9% were found for the formaldehyde-reinforced foams [16]. A previous study showed generally higher cells dimensions (81 to 428 μm) and orthotropicity (1.46 to 1.99) [6]. However, by systematically comparing these foams, we can state that the foams produced with diethylether (standard and nitric acid) present a sensibly lower density and bigger diameter. As expected, the skeletal density (1380 kg/m^3^) was similar for all the foams, with a consequently slightly higher porosity and orthotropicity for the surfactant-free foams. Generally, the foams with the surfactant presented more homogeneous cells dimensions, resulting in a more homogeneous material.

The computed tomography analysis highlighted a sensibly different morphology for the foams. In Figure 2, the structure of the cell wall is depicted for the standard foam (SA-DE), for the sulfur-free one (NA-DE), and for the one produced with aliphatic blowing agent (SA-Pent).

The features presented in Table 2 are confirmed by the CT analysis, so the SA-DE and the NA-DE are very similar: they exhibit very high porosity, lower pore homogeneity, and larger pore diameter than SA-Pent foams. Indeed, the foam formulated with aliphatic blowing agents is denser, characterized by more homogeneous and smaller pores. The porosity of the volumes represented in Figure 2 was calculated by performing a thresholding procedure, and then computing a ration between the walls and the total volume. The porosity values obtained were 95.9%, 95.4%, and 94.3%, for the standard, nitric and pentane formulations, respectively. These results are in good agreement with the experimental values of Table 2.

In view of a possible energy valorization of the tannin foams at the end of their life, their heating values were analyzed. The five foams presented similar calorific values, as reported in Table 3.

The foams showed a similar moisture content, and in particular the sulfur-free formulations (NA-DE and NA-PE) registered slightly lower values. These foams also showed slightly enhanced high heating values, which represents a further advantage for the end-life of these materials. As expected, these PCS_0_ were intermediate between the mimosa tannin (=20.25 MJ/kg), and the polyfurfuryl alcohol (=27.89 MJ/kg).

### 3.2. Intrinsic Chemical Properties

In Figure 3, the FT-IR spectra of the foams are reported.

From the chemical viewing angle, the FT-IR spectra show similar absorption bands, but the relative intensity is affected by the used catalyst. The polymers produced with sulfuric acid have more intense spectral bands in the regions between: (i) 1220 and 1150 cm^−1^ and (ii) 920 to 820 cm^−1^. The considered bands can be assigned to furanic signals [30]. In the foams catalyzed by nitric acid, a significant increment in the intensity of the bands between: (i) 1480 and 1220 cm^−1^ and (ii) 1150 and 1050 cm^−1^ are registered; both these regions are dominated by the absorption of phenolic components [31]. These observations suggest that the foams produced with nitric acid have lower furanic character than the standard foams catalyzed by sulfuric acid.

### 3.3. Extrinsic Physical Properties

The five foams were tested for their mechanical, thermal and fire resistance properties.

#### 3.3.1. Compression Strength

In Figure 4, the resistance against compression of the foams is summarized.

The compression strength registered in this study can be compared with the previous studies on tannin foams: lightweight formaldehyde-free tannin foams of 50 kg/m^3^ showed around 0.03 MPa, while heavier ones of 180 kg/m^3^ reached up to 0.45 MPa [13,22].

The foams produced with aliphatic blowing agents show higher values of compressive strength independently of the density. Generally, the mechanical resistance of tannin foams has been reported to increase by increasing their density. Nevertheless, in the present study, a major role seems to be played by the dimension of the pores. In fact, foams with smaller and more homogeneous pores have a higher compression resistance. For instance, a pentane or petrol ether foam of around 70 kg/m^3^ density (0.15 MPa) has around three times the strength of a standard or nitric acid foam with the same density (0.05 MPa).

#### 3.3.2. Thermal Conductivity

The thermal conductivity of these foams is very similar for every formulation (Figure 5), and it ranges from 33 to 41 mW/m.K. These measurements enhance the results registered in previous studies, where the thermal conductivity was between 45 and 55 mW/m.K [22,32]. In particular, the foams with aliphatic blowing agents (SA-Pent, SA-PE, and NA-PE) show significantly improved insulation capacity.

The effect of the foam density on the thermal conductivity is even more modest than the one it has on the compression strength, while the cell dimension is the dominant parameter: the smaller the cells are, the lower the thermal conductivity.

#### 3.3.3. Fire Resistance

Tannin-furanic foams generally show a self-extinguishing behavior—which was already previously described for formaldehyde-reinforced foams of similar density—which extinguished in around 80 s [6], and for alkaline tannin foams, when the self-extinguishing occurred instantaneously [33]. The fire behavior of these formaldehyde-free formulations is presented in Figure 6.

The graphic shows the mass loss of the foams during the burning and the glowing phase. After an initial 1-minute steep mass loss, due to the release of solvents, of water, and the burning of the surfaces, the foams stabilize their mass loss rate, showing their reticence for burning and never keeping the flame during glowing. All foams show a similar trend, but the NA-PE appears to be more fire sensitive by showing around a 10% increase of mass loss in the initial burning phase.

Even if the difference might be statistically less important (an average of five samples), the different behavior of the NA-PE foam can be due to the presence of solvents in some closed pores. This presence may contribute to higher mass losses due to the further feed for the flame. Even in the final part of the glowing phase, there is a slightly higher slope in the mass loss, which can be attributed to the sudden increase of oxygen due to the opening of some closed regions.

### 3.4. Extrinsic Chemical Properties

The first extrinsic chemical property considered is the percentage of acid recovery. When the foams come into contact with water, the catalyst remaining on the foam surface of the foam is leached out, and it is possible to determine the pH value of the water, and hence the percentage of acid recovery. In Figure 7, the measured pHs are reported.

It can be seen that the pH of the solutions is always similar, and stays between 2.49 and 2.75. The use of nitric acid produces a less acidic environment, which could be due to different type and concentration of the acid, to its partial evaporation, but also to the different release after leaching. Numerically, 17.8% to 23.4% nitric acid could be recovered against the 29.6% to 39.7% for sulfuric acid. These higher values can be related to the lower phenolic character of the sulfuric acid catalyzed foams (SA-DE), suggesting the weaker interaction of the catalyst, which will be released more easily.

Further, the capacity of the material to act as a filter against organic and emerging pollutants such as methylene blue and sodium dodecyl sulfate [34,35] is evaluated and reported in Figure 8.

It was observed that the foams are able to recover from 42% to 55% of methylene blue and from 15% to 27% of SDS. These results confirm the previous study of the group [21], but they highlight a marked increase in adsorption when nitric acid is used as a catalyst, carrying the absorption of methylene blue over 50% and SDS over 25%, increasing their absorption capacity.

According to the FT-IR analysis, the nitric acid-catalyzed foams seems to have a higher phenolic and a lower furanic character. Hence, the improved absorption of methylene blue and SDS with nitric acid confirms the higher importance of the tannin part in the absorption process.

## 4. Conclusions

An extended number of different formulations for the preparation of formaldehyde-free tannin foams are available, and some of them have gained industrial interest. In this work, we have presented sulfur-free foams produced using nitric acid as catalyst. They show less furanic character, less acid pH, and a higher absorption capacity for cationic and anionic pollutant by keeping the physical properties of the standard foams. Moreover, the foams obtained with the aliphatic blowing agent produce more homogeneous, smaller pores; more mechanically resistant; and are more insulating in respect to the standard ones, while maintaining the same intrinsic and extrinsic chemical properties. Finally, sulfur-free foams blown with petroleum ether have shown more homogeneous morphology, higher mechanical resistance, and lower thermal conductivity by keeping the higher phenolic character of the nitric acid catalyzed foams. Due to these properties, the NA-PE foam represents a very attractive sulfur-free product for the building insulation market, with higher chance to be industrially upscaled. Studies are currently running to further enhance the end-life opportunities for these materials.

## Figures and Tables

**Figure 1 polymers-12-00564-f001:**
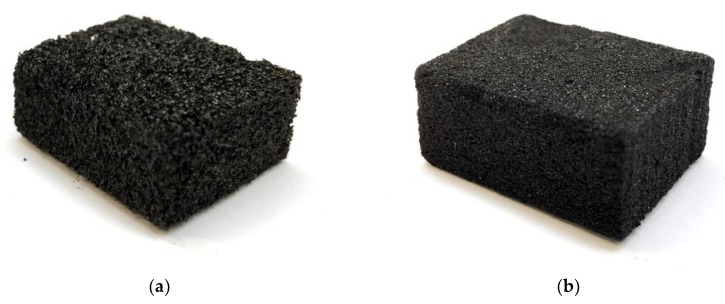
Optical appearance of the standard (SA-DE) (**a**) and high performing petroleum ether (NA-PE) foams (**b**).

**Figure 2 polymers-12-00564-f002:**
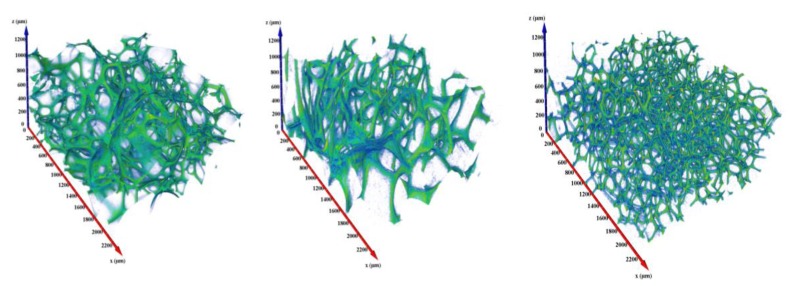
Three-dimensional Structures of the tannin foams (rendered volume ~5 mm^3^): standard, nitric acid and pentane (left to right).

**Figure 3 polymers-12-00564-f003:**
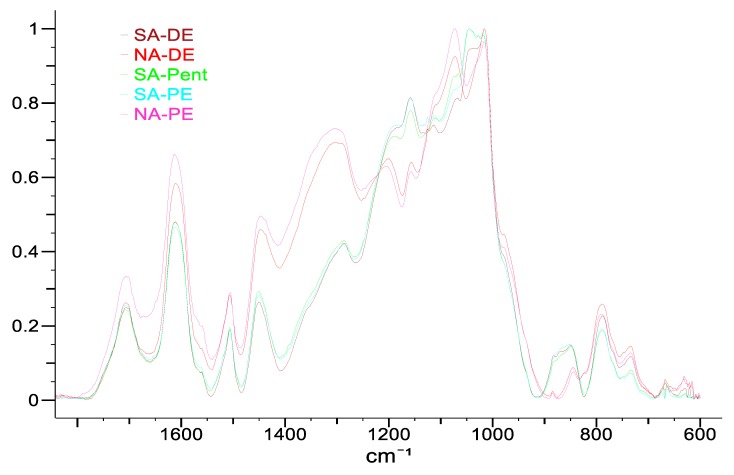
Attenuated Total Reflectance Fourier-Transform Infrared (ATR FT-IR) spectra of the five tannin-based foams between 1800 and 600 cm^−1^.

**Figure 4 polymers-12-00564-f004:**
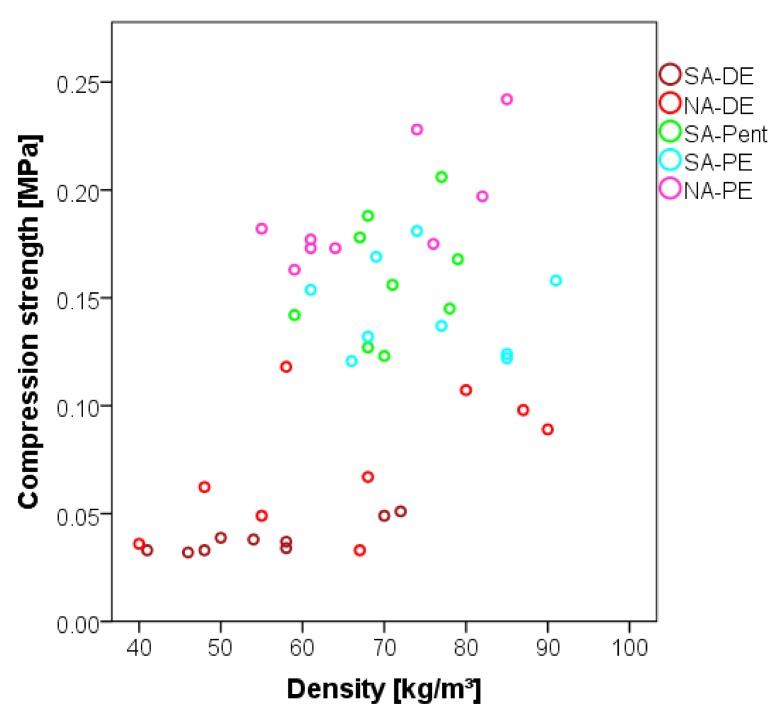
Compression strength of the foams against density.

**Figure 5 polymers-12-00564-f005:**
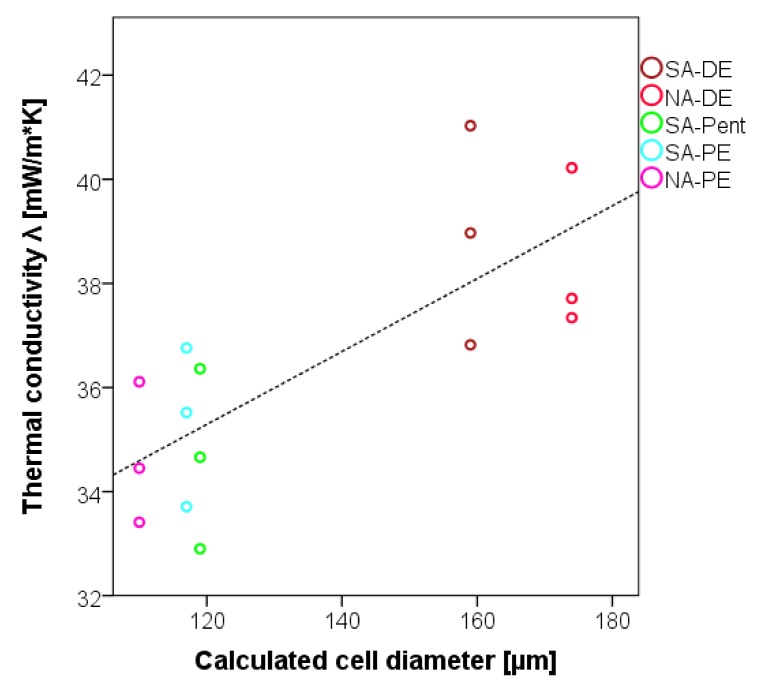
Thermal conductivity of the foams against cell diameter.

**Figure 6 polymers-12-00564-f006:**
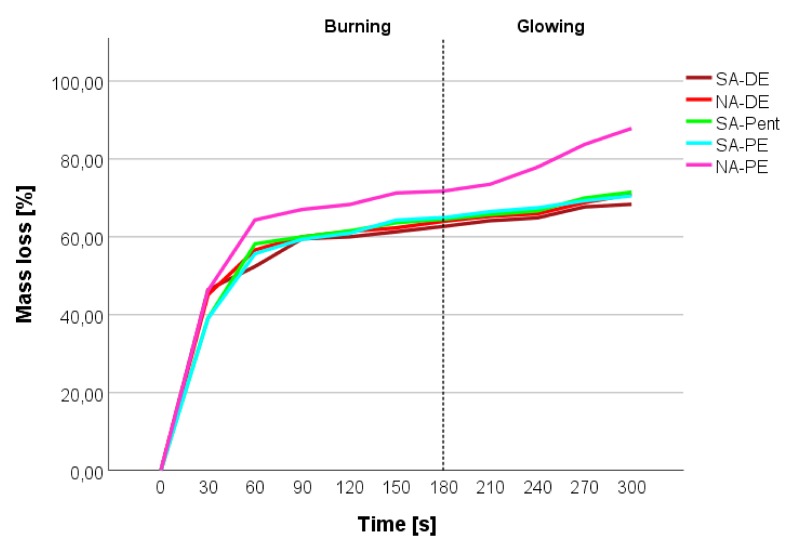
Mass loss of the tannin foams during direct flame exposure: burning and glowing phases.

**Figure 7 polymers-12-00564-f007:**
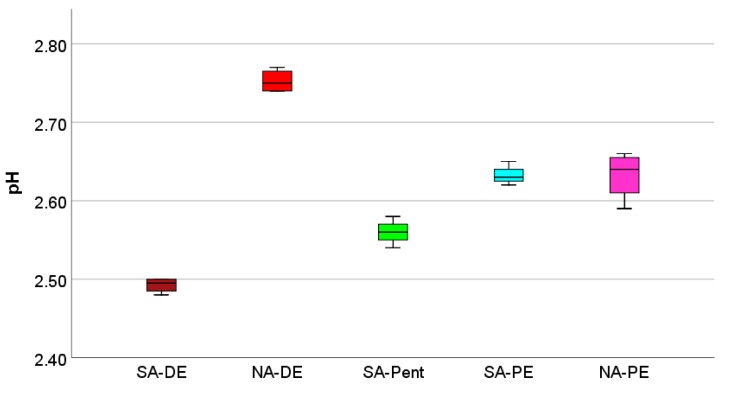
pH of the water after foam leaching.

**Figure 8 polymers-12-00564-f008:**
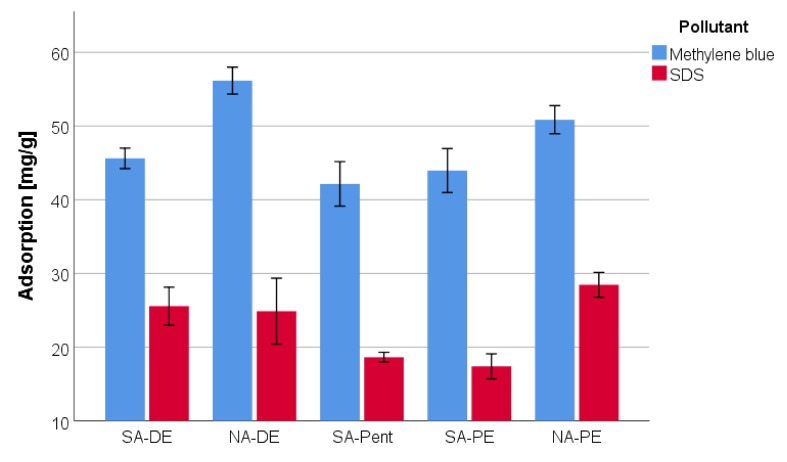
Pollutant adsorption of tannin foams.

**Table 1 polymers-12-00564-t001:** Formulation description in percentage by weight.

Formulations	Acronym	Tannin (%)	Furfuryl Alcohol (%)	Blowing Agent (%)	Tween80 (%)	Catalyst (%)
Standard	SA-DE	42.0	25.9	DEE 5.5	0.0	SA 5.8
Nitric Acid	NA-DE	42.0	25.9	DEE 5.5	0.0	NA 4.5
Pentane	SA-Pent	39.1	25.5	Pent 5.5	1.3	SA 5.7
Petroleum Ether	SA-PE	39.1	25.5	PE 5.5	1.3	SA 5.7
Nitric Acid-Petroleum ether	NA-PE	39.1	25.5	PE 5.5	1.3	NA 4.5

Note: The remaining part was water. Every formulation was prepared in tiles, five 10 × 10 × 3 cm^3^ and two 30 × 30 × 3 cm^3^.

**Table 2 polymers-12-00564-t002:** Bulk density, porosity average diameter, and orthotropicity of the tannin-based foams.

Foams	Bulk Density [kg/m^3^]	Porosity [%]	Diameter [μm]	Orthotropicity
Standard (SA-DE)	62.3 (9.6)	95.5 (0.7)	172 (55)	1.41 (0.32)
Nitric acid (NA-DE)	68.4 (11.3)	95.3 (0.8)	174 (73)	1.53 (0.32)
Pentane (SA-Pent)	82.5 (6.7)	94.0 (0.5)	118 (36)	1.30 (0.25)
Petroleum ether (SA-PE)	83.8 (9.7)	93.9 (0.7)	113 (37)	1.31 (0.26)
Nitric Acid-Petroleum ether (NA-PE)	76.3 (5.4)	93.8 (0.4)	110 (33)	1.25 (0.18)

**Table 3 polymers-12-00564-t003:** Moisture content and heating values of the powdered foams.

Foams	Moisture Content [%]	PCI_w_ [MJ/kg]	PCS_0_ [MJ/kg]
Standard (SA-DE)	12.50	18.89	22.77
Nitric acid (NA-DE)	9.69	20.30	23.57
Pentane (SA-Pent)	11.11	19.41	22.97
Petroleum ether (SA-PE)	10.55	19.22	22.61
Nitric Acid–Petroleum ether (NA-PE)	10.04	19.80	23.12

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
