# Peer review of "Synthesis and Characterization of High-Performing Sulfur-Free Tannin Foams"

_polymers, 2020, doi:10.3390/polym12030564_

Round 1

Reviewer 1 Report

Work is good, manuscript is well written. Accept

Author Response

We thank the reviewer for the appreciation. Some small modification have been done anyways, in order to fulfill the requirement of the other reviewers.

Reviewer 2 Report

  Manuscript entitled: "Synthesis and characterization
of high performing, sulfur-free tannin foam" is
well written, but requires some corrections. It is
interesting topics,
manuscript sounds scientific, there can be a lot of interest
of the results,however, the reviewer suggests making corrections:
1.authors should better describe FTIR
2.figure 3 is illegible
3.authors should better compare results with literature data
4.the novelty item is poorly described        

Author Response

We thank the reviewer for suggestions. We have modified the paper accordingly. We clarified the FTIR, we increased the dimension of fig.3 . Further we added some more comparison with previous work and we stressed on the innovation carried with this work.

Reviewer 3 Report

Comments to Authors:

Sulfur-free tannin foams were successfully produced via prepreg compression molding, the surface was treated using plasma technique, the surface morphology, mechanical properties and chemical bonding strength are well characterized.  The paper “synthesis and characterization of high performing, sulfur-free tannin foams” could be accepted after following revision.

  1. The title can be modified to “Synthesis and characterization of high performing sulfur-free tannin foams”
  2. first person pronouns should be avoid in the manuscript(abstract we)
  3. a Höfer press?
  4. For the part“3.1 Bulk density, skeletal density and porosity and 2.3.2 Cells dimension and orthotropicity” The following reference can be added to show the proper methods.

Bai, C., Franchin, G., Elsayed, H., Conte, A., & Colombo, P. (2016). High strength metakaolin-based geopolymer foams with variable macroporous structure. Journal of the European Ceramic Society, 36(16), 4243-4249.

Bai, C., Li, H., Bernardo, E., & Colombo, P. (2019). Waste-to-resource preparation of glass-containing foams from geopolymers. Ceramics International, 45(6), 7196-7202.

  1. The Intrinsic physical properties and extrinsic physical properties of matrix should be compared with previous works (other foams).
  2. The adsorption performance for the foams should be added not only for the powders.
  3. The results sulfur-free tannin foams should be compared with previous tannin foams.
  4. The optic images of foams should be given.

Author Response

Dear reviewer, dear editor:

We would like to highlight that this review may not have understood our work because we did not perform any prepreg compression molding and we did not treat any surface with plasma. We did not investigate the surface morphology, but the core one and also, we did not focus on the chemical bonding strength particularly.

However, we checked every further remark and we act as follow:

  1. We changed the title accordingly, removing the comma.
  2. We have replaced the ones used in the experimental section.
  3. Hoefer is the producer of the press. We don’t get the point.
  4. We refuse to introduce these references being them not properly related to the paper.
  5. We agree with the reviewer and we have extended this part.
  6. The adsorption performance depends on the surface exposed. In case we compare foams with different morphologies, the adsorption capacity will not give any reliable information on the adsorption capacity.
  7. This was already considered in point 5, however, the present study is made with the internal standard SA-DE which allow the systematical comparison for every property.
  8. Two pictures of the foams have been added at the beginning of the Result and discussion section.

Round 2

Reviewer 2 Report

I accept the manuscript

Reviewer 3 Report

accept as it is